# Sex and Age Influence on Association of *CYP450* Polymorphism with Midazolam Levels in Critically Ill Children

**DOI:** 10.3390/diagnostics12112797

**Published:** 2022-11-15

**Authors:** Carmen Flores-Pérez, Janett Flores-Pérez, Manuel de Jesús Castillejos López, Juan Luis Chávez-Pacheco, Karla Miroslava Tejada-Gutiérrez, Arnoldo Aquino-Gálvez, Luz María Torres-Espíndola

**Affiliations:** 1Laboratory of Pharmacology, National Institute of Pediatrics (INP), Mexico City 04530, Mexico; 2Doctorado en Ciencias Biológicas y de la Salud, División de Ciencias Biológicas y de la Salud, Universidad Autónoma Metropolitana (Doctorate in Biological and Health Sciences, Division of Biological and Health Sciences, Universidad Autónoma Metropolitana), Ciudad de México 04960, Mexico; 3Epidemiological Surveillance Unit, National Institute of Respiratory Diseases, Mexico City 14080, Mexico; 4Molecular Biology Laboratory, National Institute of Respiratory Diseases, Mexico City 14080, Mexico

**Keywords:** midazolam, cytochrome P-450 CYP3A, sex, pharmacokinetics, critically ill, pediatrics

## Abstract

Midazolam is a drug that is metabolized by cytochrome P450 (CYP450) enzymes, particularly CYP3A4 and CYP3A5. The present study aimed to determine the sex and age influence on association of CYP450 polymorphism with midazolam levels in critically ill children. Seventy-two DNA samples were genotyped by real-time PCR. Children ≤ five years of age who carry the rs776746 (T) allele in *CYP3A5* gene were associated with lower plasma midazolam levels. The concentration median in patients was 0.0 ng/mL, while in patients with the normal (C) allele, it was 438.17 ng/mL (Q_25_ 135.75–Q_75_ 580.24), *p* = 0.005. The midazolam plasmatic concentration in female patients with the minor (T) allele was 0.0 ng/mL (Q_25_0.00–Q_75_204.3), while in patients with the normal (C) allele median it was 459.0 ng/mL (Q_25_296.9–Q_75_789.7), *p* = 0.002. Analysis of the dominant model for the rs2740574 variant in *CYP3A4* revealed a median of 0.38 L/kg (Q_25_0.02–Q_75_1.5) for the volume of distribution parameter in female patients with the normal T allele, while female patients with the minor C allele showed a median of 18.1 L/kg (Q_25_7.5–Q_75_28.7) *p* = 0.02. Our results suggest an altered midazolam metabolism due to the presence the allelic rs2740574 variants of *CYP3A4* and rs776746 of *CYP3A5*, and also the strong influence of age and sex.

## 1. Introduction

Midazolam is a benzodiazepine used at the Pediatric Intensive Care Units, from premedication to maintenance of general anesthesia, mainly for effects such as sedation, hypnosis, decreased anxiety, and anterograde amnesia during diagnostic or therapeutic procedures [1,2]. Midazolam is mainly a substrate of CYP3A, CYP3A4, and CYP3A5. Regarding *CYP3A4*1.001* and *CYP3A5*3.001* polymorphisms (legacy *CYP3A5*3* and *CYP3A4*1B*, respectively), one study reported no significant differences in the pharmacokinetic parameters of midazolam between Caucasian and African American healthy volunteers [3], however, a recent study in Mexican critical pediatric patients found that patients carrying the allelic rs776746 variant in *CYP3A5* had increased plasma midazolam levels three hours after infusion discontinuation and greater drug clearance [4]. In addition, there are important pharmacokinetic modifications, particularly on drug clearance, due to factors like age, weight, illness state, ethnicity, or genotype [5]. Due to the observed differences in the CYP3A4 expression and activity in the liver and intestine within different age groups, midazolam clearance is lower in children than in adults [6]. The time needed to achieve the clinical effect on children is higher for midazolam than for any other sedative agent [7]. Physiologic differences between males and females can explain the pharmacokinetic variations, which have been described widely [8,9]. Most of the studies have failed to find significant sex differences regarding midazolam metabolism [10,11]. Except for greater clearance in females [12,13], a small sample size was considered. Although it has been proposed that sex differences in the activity of CYP3A4 exist [14,15], current evidence supporting this is limited, non-existent [16], or contradictory [17]. In previous studies of our group, the allelic variants of the *CYP450* genes (nine probes in the *CYP2B6, CYP2C9, CYP3A4,* and *CYP3A5* genes) have been analyzed, and an association of toxicity and response to treatment was reported in patients with solid tumors with the allelic *CYP3A4* and *CYP3A5* variants [18,19], so it was decided to explore them in a cohort of critically ill pediatric patients. Due to the reasons stated before, the aim of the present study was to determine the sex and age influence on association of *CYP450* polymorphism with midazolam levels in critically ill children.

## 2. Materials and Methods

### 2.1. Patients

This study was approved with number INP-068/2014 by the Research, Biosafety, and Ethics Committee (IRB 00008064) of the National Institute of Pediatrics. This cohort is special because all patients requiring to be treated in both units during the study period were included. All informed consent forms were signed previously by parents or legal guardians.

Pediatric patients 1–17 years old, of any sex, treated in the Pediatric Intensive Care Unit and Emergency Unit and sedation with midazolam through continuous intravenous (IV) infusion, starting at a 200 µg/kg/h (range 200 to 1200 µg/kg/h) rate and then adjusted according to the sedation level during hospitalization, were included. Patients with suspected hepatic or renal failure, who had undergone partial hepatectomy or partial or total nephrectomy, multiple organ failure, hypersensitivity to benzodiazepines, who had been treated with CYP3A4 inhibitors such as omeprazole, erythromycin, or clarithromycin, with blood transfusion during the last 6 months, who had suffered septic shock, or had any type of leukemia, were excluded. The elimination criteria were: incomplete sampling, serious adverse effects during treatment, and acute renal or hepatic dysfunction developed during the study.

Concomitant medications were recorded during the hospital stay and classified as CYP3A4 substrates, inducers, or inhibitors.

### 2.2. Peripheral Blood and Plasma Samples

Peripheral blood samples (3 mL) were collected in EDTA Vacutainer tubes (Becton Dickinson, Franklin Lakes, NJ, USA) and centrifuged for lymphocyte isolation and further DNA extraction. For the pharmacological analysis, three blood samples (3 mL each) were obtained from every patient. The first sample was taken once the treatment with generic midazolam (PiSA Laboratories, Mexico City, Mexico) was started at an initial IV infusion dose of 200 μg/kg/h, and was collected in a heparin Vacutainer tube (Becton Dickinson, Franklin Lakes, NJ, USA). The next two samples were collected at three and twenty-four hours after the end of treatment. All samples were centrifuged for 10 min at 800× *g* (Z326K, Hermle, Gosheim, Germany), and plasma was separated and placed in microcentrifuge tubes, labeled, and stored at −80 °C (MDF-U76VC, Panasonic, Osaka, Japan) until analysis.

### 2.3. Genotyping

Genomic DNA was isolated from peripheral blood leukocytes (n = 72) using the QIAmp DNA Mini kit (Qiagen, Hilden, Germany). For genotyping the allelic variants, rs2740574 in *CYP3A4* and rs776746 in *CYP3A5*. Genotype analysis was performed by allelic discrimination assay using TaqMan probes. For each allelic discrimination assay, specific probes labeled with different fluorophores at the 5’ end, VIC for allele 1 and FAM for allele 2 were obtained from Applied Biosystems, with both probes characterized by having a “quencher” (TAMRA) at the 3’ end which, while the probe remains intact, inhibits fluorescence emission, and a passive signal (ROX) will be used. The PCR reaction was carried out as follows in an microtube, and the following was added for 48 samples: 240 μL of TaqMan PCR Master Mix 2X, 24 μL of 20X of each probe (Applied Biosystems, Foster City, CA, USA), and 120 μL of nuclease-free water (Ultra Pure^TM^) DEPC treated water, Invitrogen, Carlsbad, CA, USA. Fluorescence for each sample was quantified in a StepOne using SDS 2.2.1 software for allelic discrimination (Applied Biosystems Foster City, CA, USA).

Finally, we analyzed the SNPs based on inheritance models: dominant (mutant homozygous plus heterozygous versus normal homozygous) and recessive (mutant homozygous versus heterozygous plus normal homozygous), by sex (male and female), and age (≤five and >five years old) stratification.

### 2.4. Quantification of Midazolam Plasma Levels

The method is simple and reliable by high-performance liquid chromatography (HPLC) and requires only a small volume of plasma (200 µL), which has been previously validated and published by our group [20].

### 2.5. Validation Criteria

Analytical validation met the following criteria: selectivity, linearity, reproducibility, repeatability, quantification limit, and stability to different conditions according to Official Mexican Standard NOM-177-SSA1, 2013 [20], which were under the FDA and EMA international guideline for bioanalytical method validation [21,22].

### 2.6. Reagents

All reagents were analytical or HPLC grade. Midazolam hydrochloride, acetonitrile, and diethyl ether were obtained from Sigma (St. Louis, MO, USA), propranolol from ICN Biomedicals Inc. (Aurora, OH, USA), potassium dihydrogen phosphate and sodium hydroxide from Merck (Darmstadt, Germany), and deionized water was obtained from Elix 35/Milli Q water purification system (Millipore, Molsheim, France).

### 2.7. Equipment and Chromatographic Conditions

A Waters (Milford, MA, USA) chromatographic system with a 515 pump, a 717 autosampler, a 2487 UV detector, and a degasser (Metachem, Torrance, CA, USA) were used. Analytical separation was realized in a reversed-phase column Pursuit C18 (5 µm, 150 × 3.9 mm i.d., Agilent, Santa Clara, CA, USA) maintained at room temperature (25 °C), while the autosampler was kept at 4 °C. The mobile phase was 35 mM phosphate buffer solution pH 4.4 (adjusted with phosphoric acid) and acetonitrile (70:30 *v*/*v*) with an isocratic flow rate (1 mL/min); absorbance was monitored at 220 nm. Data were obtained and processed using Empower Pro™ version 2 software (Waters Inc., Milford, MA, USA).

### 2.8. Extraction Procedure

Midazolam extraction was done with the addition of 3 mL of diethyl ether to 200 μL of plasma (alkalinized with 100 μL of a 1 N sodium hydroxide solution). The solution was then vortexed for 1 min and centrifuged at 800× *g* for 10 min. To ease the separation of the organic phase, the tubes were frozen at −80 °C for 10 min and then the organic layer was decanted to another tube and evaporated at 40 °C under airstream. The dry residue was dissolved in a mixture of 190 μL of mobile phase and 10 μL of 50 μg/mL propranolol solution as an external standard. Then, 100 μL aliquots were injected into the chromatographic system.

### 2.9. Determination of Pharmacokinetic Parameters

The data were adjusted to one-compartment pharmacokinetic model for calculating the pharmacokinetic parameters and the following formulae were considered:

Elimination constant (ke) = lnCp2 − ln Cp1/t2 − t1; elimination half-life (t1/2) = ln2/ke; volume of distribution (Vd) = D/Cp0 and clearance (CL) = ke × Vd; where Cp is the plasma concentration, t denotes time, D is the dose administered, and Cp0 is the plasma concentration at zero time [23,24].

Once the pharmacokinetic parameters were obtained, stratification by sex (male and female) and age (≤five and >five years) was performed for statistical analysis.

### 2.10. Statistical Analysis

The median and the interquartile range (Q_25_–Q_75_) of the midazolam concentrations at three- and twenty-four hours post-infusion were compared between the allelic variants, stratified by sex and age. The Mann–Whitney U test between age, sex groups, and comparison between the presence of allelic variants and midazolam levels was performed. For the comparison of plasma levels of the drug at three- and twenty-four hours, the Wilcoxon test was used. A *p*-value < 0.05 indicated a statistically significant association. Variables were stratified by sex (male and female) and age (≤five and >five years old).

For the statistical analysis, SPSS v20.0 (Statistical Package for the Social Sciences, IBM Corp., Armonk, NY, USA) was used.

## 3. Results

### 3.1. Study Population Characteristics

Eighty-one patients were invited to participate in the study, of whom 72 accepted and met the study criteria and were included by means of an informed consent that they signed. Forty-three of these patients were male and twenty-nine were female. The demographic and clinical data are shown in Table 1.

The genotyping analysis was performed on all 72 patients (data not shown), while, for the determination of pharmacokinetic parameters, nine patients were excluded because their clinical status was compromised due to organ failure, seven subjects were eliminated due to death, and one participant did not have a sufficient blood sample, leaving a total of 55 patients, of whom 21 were female and 34 were male.

### 3.2. Effect of Midazolam Concentrations according to the CYP3A4 and CYP3A5 Genotypes Stratified by Age

The genotype association’s magnitude with the midazolam plasmatic levels was determined through a comparison of the medians of the dominant and recessive inheritance models for both allelic variants (rs2740574 in *CYP3A4* and rs776746 in *CYP3A5*) stratified by age (≤five and >five years old).

Analysis of the dominant inheritance model showed a statistically significant association for TT+TC in the rs776746 variant in children ≤five years old. After three hours, the midazolam plasma concentration median in patients with normal CC (major) genotype was 438.17 ng/mL (Q_25_ 135.75–Q_75_ 580.24), while in patients with the TT+TC (minor), the genotype was 0.0 ng/mL (Q_25_0.00–Q_75_0.00), *p* = 0.005 (Table 2).

After twenty-four hours, patients with the major genotype had a concentration of 0.00 ng/mL (Q_25_0.00–Q_75_196.64), and patients with the minor genotype had a midazolam concentration median of 0.00 ng/mL (Q_25_0.00–Q_75_0.00), *p* = 0.5. In children older than five years old, no statistically significant association was found.

The recessive inheritance model analysis of the rs2740574 variant showed no association between its presence, midazolam levels, and age (Table 2). No association between the allelic variant, pharmacokinetic parameters, and age was found after analysis with both inheritance models (data not shown).

### 3.3. Effect of Midazolam Concentrations according to the CYP3A4 and CYP3A5 Genotypes Stratified by Sex

The medians of the dominant and recessive inheritance models stratified by (male and female) for both allelic variants (rs2740574 and rs776746) were compared to obtain the genotype association’s magnitude with the midazolam plasmatic levels.

A statistically significant association was found in females with the TT+TC of the rs776746 variant in the dominant inheritance model analysis. After three hours, a median of 459.0 ng/mL (Q_25_296.9–Q_75_789.7) for the midazolam plasmatic concentration was found in female patients with the major genotype, while in patients with the minor genotype it was 0.0 ng/mL (Q_25_0.00–Q_75_204.3), *p* = 0.002 (Table 3).

Twenty-four hours post-infusion, female patients with the major genotype showed a concentration of 0.00 ng/mL (Q_25_0.00–Q_75_171.6), and patients with the minor genotype had a midazolam concentration median of 0.00 ng/mL (Q_25_0.00–Q_75_79.0), *p* = 0.3. In male patients, no association was found with either allelic variant (Table 3).

### 3.4. Effect of CYP3A4 and CYP3A5 Genotypes over the Pharmacokinetic Parameters Stratified by Sex

To determine the genotype association’s magnitude with the midazolam pharmacokinetic parameters, the medians were compared using the dominant and recessive inheritance models for both allelic variants (rs2740574 and rs776746), stratifying by sex (male and female).

Analysis of the dominant model for the rs2740574 variant revealed a median of 0.38 L/kg (Q_25_0.02–Q_75_1.5) for the Vd parameter in female patients with the TT (major) genotype. Meanwhile, female patients with the CC+CT (minor) genotype showed a median of 18.1 L/kg (Q_25_7.5–Q_75_28.7) *p* = 0.02 (Table 4). The dominant model in males found no statistically significant association.

The recessive inheritance model analysis for the rs2740574 variant presented no association among its presence, pharmacokinetic parameters, or sex. This was also the case for the rs776746 variant for both inheritance models between the allelic variant presence, pharmacokinetic parameters, and sex, as no association was found (Table 4).

## 4. Discussion

In the absence of pediatric pharmacokinetic studies to guide the safe and effective use of drugs, pediatric dosing can be directed by knowing anatomic and physiologic factors which help to understand drug disposition in this population [25]. Midazolam may be more effective in children because it has a faster onset of action and shorter duration than other benzodiazepines [26]. Drug disposition in children differs from that of adults due to age, absorption, distribution, metabolism, and excretion [27]. These differences may be due to a smaller intestine and altered permeability; in addition to gastric emptying time, the intestinal transit time, bile fluid production, and blood flow to the intestines and liver may be modified in children [28].

Individual response to drugs is highly variable. It can be regulated by genetic and environmental factors, age, sex, body size, ethnicity, diseases, kidney and liver pathologies, diabetes, and obesity [29].

Females and males may respond differently to the same therapeutic regimen due to sex-specific variations in pharmacokinetics and pharmacodynamic profiles [30,31,32]. The expression and activity of drug-metabolizing CYP450 enzymes can be affected by many factors, including genetic polymorphisms and sex, leading to changes in drug metabolism and therapeutic effects [33,34,35].

It is estimated that in human beings there is a 40% pharmacokinetic difference between men and women; women are smaller, have higher body fat content, have less muscle compared to men, and their total body water is lower (~15–20%) [32].

Sex-related differences in pharmacokinetics and pharmacodynamics include noticeable physiologically related differences, such as body fat content and hormonal influence, among others [29]. The possibility of variations in the menstrual cycle and the renal, cardiovascular, and hematological systems, which may affect protein binding and volume of distribution, has been studied [30]. Although estrogen appears to affect CYP3A activity in vitro, no menstrual cycle changes have been found in the metabolism of CYP3A substrates in vivo [11,36]. Regarding *CYP3A5*3* and *CYP3A4*1B* polymorphisms, Miao et al. (2009) [3] reported that there were no significant differences in the pharmacokinetic parameters of midazolam. In contrast, we found a sex-related influence on the concentrations and volume of distribution of midazolam (Table 3 and Table 4, respectively), where it is shown that females who carry the *CYP3A5* rs776746 variant comparing to girls with the normal allele, showing a decrease in plasma concentrations since three hours after the end of treatment and an increase in the volume of distribution females who carry the *CYP3A4* rs2740574 variant comparing to girls with the normal allele.

In women, the volume of distribution of lipophilic drugs is increased [8,37,38], including benzodiazepines such as diazepam [34,35] and midazolam [10]. In the present work, we found that in the dominant model, patients carrying the *CYP3A4* rs2740574 (CC+CT) variant had a larger volume of distribution compared to patients with the normal genotype (TT).

The same dose of a lipophilic drug will have a lower serum concentration in a female compared to a male with the same body weight because there is a relatively larger lipophilic compartment in which the drug resides. In our study, we observed that in the dominant model, female patients with the *CYP3A5* rs776746 variant (TT+TC) did not show plasma levels of the drug compared to patients with the normal genotype (CC) three hours after the end of treatment.

Most studies have found no significant sex differences in midazolam metabolism [10,11,39,40], except for increased clearance in females [12,13], although this was a small sample size. A study by Wolbold et al. (2003) [41] found that the CYP3A4 mRNA levels were twice as high in the liver of women than in men. A greater clearance mediated by CYP3A4 has been reported in women compared to men [42,43]. Likewise, some reports have proposed to find the sex differences regarding the pharmacokinetic parameters of the CYP3A4 or CYP3A5 substrates; a study realized by Chen et al. (2006) [42], in which sex differences of the midazolam blood concentrations in adults were reported, concluded that women had a higher liver and intestinal enzymatic activity of CYP3A compared to men. Nonetheless, studies showing sex differences with the *CYP3A5* variant in children were not found.

Studies have already been realized in adults evaluating the effect of age-related blood and plasma drug concentrations, with contradictory results, however. From seven studies conducted with intravenous midazolam, only five evaluated the age, and, just in one, with men participating, a significantly lower midazolam clearance was found as the age increased [10]. However, one study found women’s clearance decreases as age increases [44], and another two, who tested intravenous midazolam, found no significant differences with age [10,45]. On the other hand, the intravenous use of midazolam without considering sex as a variable for the analysis has been reported, and the age differences in clearance were not significant [46].

This is the first study to show differences in midazolam plasma concentrations in critically ill patients ≤5 years old who carry the *CYP3A5* rs776746 variant; compared to children with the normal allele, they showed a decrease in plasma concentrations from three hours after the end of treatment, inferring an increase in clearance of the drug, and could require more doses of the drug to remain sedated, and also we could not find age-related differences in clearance because of the small sample size in this group. Some reports mentioned that the pharmacokinetics of xenobiotics could differ widely between children and adults due to physiological differences, metabolism, protein binding, immaturity of enzyme systems, the volume of distribution, and clearance mechanisms [47,48,49,50].

Among the limitations of the study was the small sample size, and death and organ failure during follow-up, which made it difficult to determine the pharmacokinetic parameters. Possible adverse reactions with concomitant medications were not reported in the clinical records, and no information was found on the use of assessment scales in this population.

This is a helpful strategy, since the results obtained could lay the foundations for the design of personalized therapeutic schemes, resulting in the best cost–benefit and monitoring of adverse events for different drugs metabolized by CYP3A in the Mexican pediatric and probably in the Latin American population.

## 5. Conclusions

Our results suggest an altered midazolam metabolism due to the presence of the allelic rs2740574 variants of *CYP3A4* and rs776746 of *CYP3A5,* and the strong influence of age and sex.

In critically ill patients ≤5 years old carrying the allelic rs776746 variant in *CYP3A5*, the plasma concentrations of midazolam decreased at three hours after the end of treatment compared to children with the normal allele. They may require more doses of midazolam to remain sedated.

Females who carry the *CYP3A5* rs776746 variant had decreased plasma concentrations three hours after the end of treatment compared to girls with the normal allele, and also showed an increase in the volume of distribution in females who carry the *CYP3A4* rs2740574 variant compared to those with the normal allele, and likely could need adjustment of doses of midazolam to achieve the desired effect.

Further studies in larger samples are required to validate these findings and the relevance of these variants in *CYP3A4* and *CYP3A5* genes.

## Figures and Tables

**Table 1 diagnostics-12-02797-t001:** Demographic and clinical data.

Characteristic	Total N (%)
Number of patients	72 (100)
Sex (male/female)	43/29 (59.7/40.3)
Age (years) *	8.5 (1–17)
Weight (kg) * Midazolam dose (μg/kg/h) *	21.55 (4.7–85) 200 (50–1200)
Main diagnostic CNS tumor Community-acquired pneumonia Epilepsy/Respiratory insufficiency Other	13 (18.06) 11 (15.3) 8 (11.1) 32 (44.5)
Main concomitant drugs Paracetamol ^a^ Buprenorphine ^b^ Omeprazole ^a^ Dexamethasone ^b,c^ Fentanyl ^b^	57 (79) 49 (68) 44 (61) 22 (30) 19 (26)

* Data are expressed in median (range); metabolism of CYP3A4: a, minor substrate; b, major substrate; c, weak inducer.

**Table 2 diagnostics-12-02797-t002:** Midazolam concentration according to genotype and stratified by age.

Age (years)	Genotype	N (%) 55 (100)	Midazolam at 3 h	*p*-Value	Midazolam at 24 h	*p*-Value
			Median (Q_25_–Q_75_)		Median (Q_25_–Q_75_)	
	*CYP3A4*/rs2740574					
≤five	Dominant					
	CC+CT	2 (9)	386.61 (191.98–581.24)	0.2	111.49 (0.0–222.9)	0.9
	TT	21 (91)	0.0 (0.0–459)		0.0 (0.0–0.0)	
	Recessive					
	CC	0	NC	NC	NC	NC
	CT+TT	23 (100)	79.52 (0.00–487.96)		1.0 (0.00–85.14)	
>five	Dominant					
	CC+CT	5 (16)	23.51 (0.00–96.2)	0.1	0.00 (0.00–79.03)	0.6
	TT	27 (84)	146.83 (0.00–332.5)		0.00 (0.00–133.23)	
	Recessive					
	CC	0	NC		NC	
	CT+TT	32 (100)	119.67 (0.00–332.5)	0.3	0.00(0.00–133.23)	0.5
	*CYP3A5*/rs776746					
≤five	Dominant					
	TT+TC	11 (48)	0.0 (0.00–0.00)	0.005 *	0.00 (0.00–0.00)	0.5
	CC	12 (52)	438.17(135.75–580.24)		0.00 (0.00–196.64)	
	Recessive					
	TT	0	NC	NC	NC	NC
	TC+CC	23 (100)	79.52 (0.00–487.96)		0.00 (0.00–85.14)	
>five	Dominant					
	TT+TC	16 (50)	27.45 (0.00–206.96)	0.055	0.00 (0.00–89.09)	0.2
	CC	16 (50)	276.04(15.05–370.29)		62.97 (0.00–172.0)	
	Recessive					
	TT	2 (6)	104.7 (0.00–209.5)	0.6	84.46 (0.00–168.9)	0.8
	TC+CC	30(94)	119.67 (0.00–342.5)		0.00 (0.00–99.1)	

* Statistical significance was calculated using the nonparametric Mann–Whitney U test; NC = Not calculable.

**Table 3 diagnostics-12-02797-t003:** Midazolam concentration according to the genotype and stratified by sex.

Sex	Genotype	N (%) 55 (100)	Midazolam at 3 h	*p*-Value	Midazolam at 24 h	*p*-Value
			Median (Q_25_–Q_75_)		Median (Q_25_–Q_75_)	
	*CYP3A4*/rs2740574					
	Dominant					
Male	CC+CT	6 (17.6)	107.7 (0.00–342.6)	0.9	0.00 (0.00–195.2)	0.7
	TT	28 (82.4)	27.4 (0.00–301.3)		0.00 (0.00–133.9)	
Female	CC+CT	1 (5)	NC	0.3	NC	1
	TT	20 (95)	288.4 (0.00–563.6)		0.00 (0.00–133.2)	
	Recessive					
Male	CC	0	NC	0.4	NC	0.6
	CT+TT	34 (100)	39.21 (0.00–322.5)		0.00 (0.00–168.9)	
Female	CC	0	NC	NC	NC	NC
	CT+TT	21 (100)	275.6 (0.00–548.0)		0.00 (0.00–99.1)	
	*CYP3A5*/rs776746					
	Dominant					
Male	TT+TC	17 (50)	0.00 (0.00–146.8)	0.07	0.00 (0.00–0.00)	0.4
	CC	17 (50)	140.3 (0.00–342.6)		0.00 (0.00–171.0)	
Female	TT+TC	10 (47.6)	0.00 (0.00–204.3)	0.002 *	0.00 (0.00–79.0)	0.3
	CC	11 (52.4)	459.0 (296.9–789.7)		0.00 (0.00–171.6)	
	Recessive					
Male	TT	2 (6)	104.7 (0.00–209.5)	0.9	84.4 (0.00–168.9)	0.9
	TC+CC	32 (94)	39.2 (0.00–332.5)		0.00 (0.00–135.0)	
Female	TT	0	NC	NC	NC	NC
	TC+CC	21 (100)	275.6 (0.00–548.0)		0.00 (0.00–99.1)	

* Statistical significance was calculated using the nonparametric Mann–Whitney U test; NC = Not calculable.

**Table 4 diagnostics-12-02797-t004:** Effect of the allelic variant on the pharmacokinetic parameters of midazolam stratified by sex.

Sex	Allelic	ke (h^−1^)	*p*-Value	t 1/2 (h)	*p*-Value	Vd (L/kg)	*p*-Value	CL (L/kg/h)	*p*-Value
	Variant	Median (Q_25_–Q_75_)		Median (Q_25_–Q_75_)		Median (Q_25_–Q_75_)		Median (Q_25_–Q_75_)	
	*CYP3A4*/ rs2740574								
	Dominant								
Male	CC+CT	0.026 (0.12–0.008)	0.7	25.2 (5.5–25.7)	0.7	5.8 (0.2–6.2)	0.9	0.11 (0.03–0.16)	0.5
	TT	0.046 (0.20–0.01)		15.0 (3.3–35.5)		3.3 (0.5–10.5)		0.13 (0.02–0.38)	
Female	CC+CT	0.04 (0.08–0.07)	0.5	53.7 (8.45–99)	0.4	18.1 (7.5–28.7)	0.02 *	1.2 (0.05–2.35)	0.1
	TT	0.05 (0.1–0.02)		10.9 (6.5–34.6)		0.38 (0.02–1.5)		0.01 (0.001–0.06)	
	Recessive								
Male	CC	0	NC	0	NC	0	NC	0	NC
	CT+TT	0.03 (0.2–0.01)		16.2 (3.3–35.6)		3.5 (0.2–9.8)		0.10 (0.02–0.3)	
Female	CC	0	NC	0	NC	0	NC	0	NC
	CT+TT	0.05 (0.08–0.02)		10.9 (6.6–42.1)		0.5 (0.03–1.7)		0.02 (0.001–0.06)	
	*CYP3A5*/ rs776746								
	Dominant								
Male	TT+TC	0.078 (0.25–0.018)	0.1	8.8 (2.7–36.4)	0.3	3.3 (0.8–18.0)	0.7	0.16 (0.03–0.45)	0.9
	CC	0.33 (0.10–0.01)		21.0 (7.4–55.4)		4.0 (0.2–9.6)		0.08 (0.015–0.23)	
Female	TT+TC	0.06 (0.1–0.02)	0.4	9.03 (2.6–18.2)	0.3	1.00 (0.01–8.40)	0.9	0.07 (0.001–0.105)	0.6
	CC	0.05 (0.08–0.1)		12.3 (7.9–42.07)		0.49 (0.05–1.35)		0.018 (0.002–0.049)	
	Recessive								
Male	TT	0.11 (0.2–0.009)	0.6	40.1 (3.2–77.0)	0.7	9.0 (0.00–18.07)	0.8	0.08 (0.00–0.16)	0.8
	TC+CC	0.038 (0.16–0.01)		17.7 (4.2–45.9)		3.6 (0.5–10.7)		0.13 (0.02–0.42)	
Female	TT	0	NC	0	NC	0	NC	0	NC
	TC+CC	0.05 (0.08–0.02)		12.1 (6.8–34.6)		0.5 (0.05–1.8)		0.023 (0.001–0.09)	

* Statistical significance was calculated using the nonparametric Mann–Whitney U test; NC = Not calculable.

## Data Availability

Not applicable.

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
