# Peer review of "Sex and Age Influence on Association of CYP450 Polymorphism with Midazolam Levels in Critically Ill Children"

_diagnostics, 2022, doi:10.3390/diagnostics12112797_

Round 1
Reviewer 1 Report
The article is aimed to determine the sex and age influence on the association of CYP450 polymorphism with midazolam levels in critically ill children. The title is “Sex and Age Influence on Association of CYP450 Polymorphism with Midazolam Levels in Critically Ill Children”.
1. The sample size of the study is relatively small.
2. Several factors influence the outcome of the study. Please discuss these.
3. The discussion part is too short. Please review the literature and add more details in the discussion section.
4. Please add the limitations of the study.
5. What is the new knowledge of the report?
6. Please recommend to the readers “How to apply this knowledge?”.
Author Response
"Please see the attachment"

Reviewer 2 Report
The authors previously published influence of rs776746 (CYP3A5) and 2740574 (CYP3A4) on midazolam metabolism on the same cohort (ref #4, https://onlinelibrary.wiley.com/doi/full/10.1111/jcpt.13388). In this manuscript, the authors specifically measure the effects of age and sex at birth.
· Transparency is recommended. When citing study within Introduction, authors should acknowledge it is their work. When describing Methods, authors might indicate methods were previously published then provide a brief overview and highlight what is specific to the analysis performed to address aim of the age, sex study.
Table 1 reflects cohort of 72 children. Section 3.2 notes values for 58 children were obtained but Tables 2 & 3 reflect comparisons among 55. Section 3.2 needs to be corrected or clarified.
Assertions in Discussion and Conclusion need to be clarified. Influence of age and CYP3A4 was not demonstrated – only PK effect was Vd differences associated with CYP3A4 in females (Table 4).
The CYP3A4 nomenclature is confusing as *1B is now referred to as *1.001 with variant (-392G>A or C>T (minus strain)) being CYP3A4*1.002 (legacy CYP3A4*1A) (https://www.pharmgkb.org/vip/PA166169915). Perhaps authors can clarify in the Introduction that they are using legacy nomenclature.
Author Response
"Please see the attachment"
